# Development platform for artificial pancreas algorithms

**Mohamed Raef Smaoui** **[1]\*, Remi Rabasa-Lhoret[2,3], Ahmad Haidar[4]**

**1** Computer Science Department, Faculty of Science, Kuwait University, Kuwait City, Kuwait, **2** Department of Nutrition, Faculty of Medicine, Université de Montréal, Montréal, Canada, **3** Institut de Recherches Cliniques de Montréal, Montréal, Canada, **4** Department of Biomedical Engineering, Faculty of Medicine, McGill University, Montreal, Canada

\* msmaoui@cs.ku.edu.kw

## Abstract

### Background and aims

Assessing algorithms of artificial pancreas systems is critical in developing automated and fault-tolerant solutions that work outside clinical settings. The development and evaluation of algorithms can be facilitated with a platform that conducts virtual clinical trials. We present in this paper a clinically validated cloud-based distributed platform that supports the development and comprehensive testing of single and dual-hormone algorithms for type 1 diabetes mellitus (T1DM).

### Methods

The platform is built on principles of object-oriented design and runs user algorithms in real-time virtual clinical trials utilizing a multi-threaded environment enabled by concurrent execution over a cloud infrastructure. The platform architecture isolates user algorithms located on personal machines from proprietary patient data running on the cloud. Users import a plugin into their algorithms (Matlab, Python, or Java) to connect to the platform. Once connected, users interact with a graphical interface to design experimental protocols for their trials. Protocols include trial duration in days, mealtimes and amounts, variability in mealtimes and amounts, carbohydrate counting errors, snacks, and onboard insulin levels.

### Results

The platform facilitates development by solving the ODE model in the cloud on large CPU-optimized machines, providing a 62% improvement in memory, speed and CPU utilization. Users can easily debug & modify code, test multiple strategies, and generate detailed clinical performance reports. We validated and integrated into the platform a glucoregulatory system of ordinary differential equations (ODEs) parameterized with clinical data to mimic the inter and intra-day variability of glucose responses of 15 T1DM patients.

**Data Availability Statement:** All relevant data are within the paper and its Supporting Information files.

**Funding:** The authors received no specific funding for this work.

**Competing interests:** The authors have declared that no competing interests exist.

## Conclusion

The platform utilizes the validated patient model to conduct virtual clinical trials for the rapid development and testing of closed-loop algorithms for T1DM.

## Introduction

Clinical trials are an integral part in the research and development process of the artificial pancreas (AP) [1, 2]. The artificial pancreas currently aims to improve glucose control and to rid T1DM patients from the continuous task of monitoring their glucose levels and manually administering insulin infusions and injections [3]. At the core of the AP technology lies a titrating algorithm responsible for computing an amount of insulin required to maintain patients' glucose in acceptable physiological levels [4, 5]. Several algorithms have been tested in clinical settings and have provided promising results to the potential efficacy of AP solutions compared to the standard open-loop treatments [4, 6–11].

The development of computer simulation environments and glucose modeling work has been critical in the evolution of AP algorithms. With the increasing availability of clinical data, T1DM patient mathematical models have continuously been improved to better approximate patient behavior and to strive to replicate clinical findings [12–17]. Among the powerful simulation environments that have been designed and made available for computational researchers to utilize in diabetes control algorithm development are the AIDA educational package [18, 19], the UVA/Padova Simulator [15] and the Cambridge Simulation Environment [12]. The latter two simulators have been validated against results of clinical studies, providing fair simulation models for AP algorithm developers [12, 20].

To be considered clinically relevant and commercially viable, algorithms designed for the AP must undergo several clinical trials and present acceptable control of patient glucose levels under normal and extreme conditions in both inpatient and outpatient settings. Algorithms must prove effective clinically in controlling post-meal glycemic excursions [21, 22] when patients eat different meals and amounts, when patients undergo exercise [23], when patients incorrectly assess carbohydrate intake [24], when patients experience mood swings and hormonal changes that alter normal glucose trends [25, 26], and when AP hardware sensor readings react abnormally [27]. Algorithms are expected to maintain glucose readings outside of hypoglycaemia ranges to avoid severe health complications [28–30]. Any improvements to existing algorithms need to remain bug-free and maintain foregoing performance.

Diabetes simulators are still far from providing rigorous environments that mimic observed clinical variabilities and are not architected to efficiently undergo thorough automated testing of different clinical protocols. Mathematical models that replicate or e-clone patients need constant improvement, and need to account for increasing physiological details and for emerging data made available by recent clinical studies [7, 31]. For example, improved models for exercise would help train algorithms on proper dosing strategies during and after exercise, and novel uses of insulin analogs along with non-insulin adjunctive therapies significantly alter glucose-insulin dynamics in T1DM patients [32–34]. Novel simulator environments need to support algorithm development in different computer languages and provide extensive clinical protocols that could be utilized and replicated to test treatments under varying conditions [35].

To improve on the current diabetes simulators in the T1DM field, we introduce in this paper a computational development platform to efficiently develop and comprehensively test novel single and dual-hormone algorithms utilizing different clinical protocols against varying

patient models. The platform assists developers in rapidly creating robust algorithms that are fault-tolerant and could be extended to work outside clinical settings. Moreover, to overcome the current architectural and simulation limitations of existing simulators, the platform in this paper has been designed as a cloud-based distributed system built with software engineering object-oriented principles to support concurrent algorithm development in multiple computer languages, support month-long simulations, and re-enforces intra-patient daily variability. The platform facilitates real-time code debugging for algorithm developers and runs on different operating systems. Addressing a significant security issue with current simulators, the platform compartmentalizes and isolates user algorithms from virtual patients. The platform aims to assist researchers and developers in the early stages of AP algorithm and testing development prior to conducting clinical trials and pursuing commercial development. Furthermore, we validate in this paper a virtual patient model against a previously conducted clinical trial to provide developers on the platform with a patient model they can directly use.

## Methods

### Platform architecture overview

The platform is comprised of several object-oriented individual programs. First, a program running a glucoregulatory system of ordinary differential equations (ODE) is responsible for modeling the glycemic responses of T1DM patients to glucose intake and insulin / glucagon delivery. Second, a controller algorithm that a user integrates into the platform through an interface plugin regulates insulin and glucagon delivery systems. Third, a program running a graphical user interface (GUI) allows a user to design and set experimental protocols for a virtual clinical trial and track outcome progress in real time. Finally, an application engine interlays the execution flow among the GUI, the glucoregulatory system, and the controller algorithm. The platform is highly scalable. While a controller algorithm runs on a user's machine, the computations of the glucoregulatory system are solved in the cloud allowing users to launch tens of simultaneous runs. This separation uplifts heavy computations from a user's machine and allows for more CPU power and memory to be allocated to the execution of longer simulation durations. The engine prepares a report summarizing the controller algorithm performance and clinically relevant outcomes for the entire virtual trial.

### Cloud infrastructure

Fig 1 provides a detailed schematic of the cloud infrastructure designed to power the development platform. Compartmentalizing the platform's elements across a cloud architecture not only allows for faster development and testing of control algorithms but also introduces a feasible way for individuals to automatically and efficiently test tens of scenarios using standard grade personal machines. The separation between client machines and the cloud service provides an additional layer of security encapsulating sensitive patient parameters from a user and securing unwarranted access to control algorithms.

The Development Platform is composed of two main parts: the client machine and the cloud service. The client machine hosts a user's closed-loop control algorithm, a graphical user interface application, and a system engine. The cloud service runs the glucoregulatory model and stores the virtual patient data (ODE parameters). Client machines communicate with the cloud service through a centralized service point, the Elastic Load Balancer (ELB). The ELB is responsible for channeling and balancing traffic to various Web Servers running the glucoregulatory model. The Web Servers are considered worker machines that listen for simulation requests and solve the ODE model iteratively. The ELB contains a protocol to scale up and launch additional Web Servers as traffic increases or scale down and remove Web Servers as

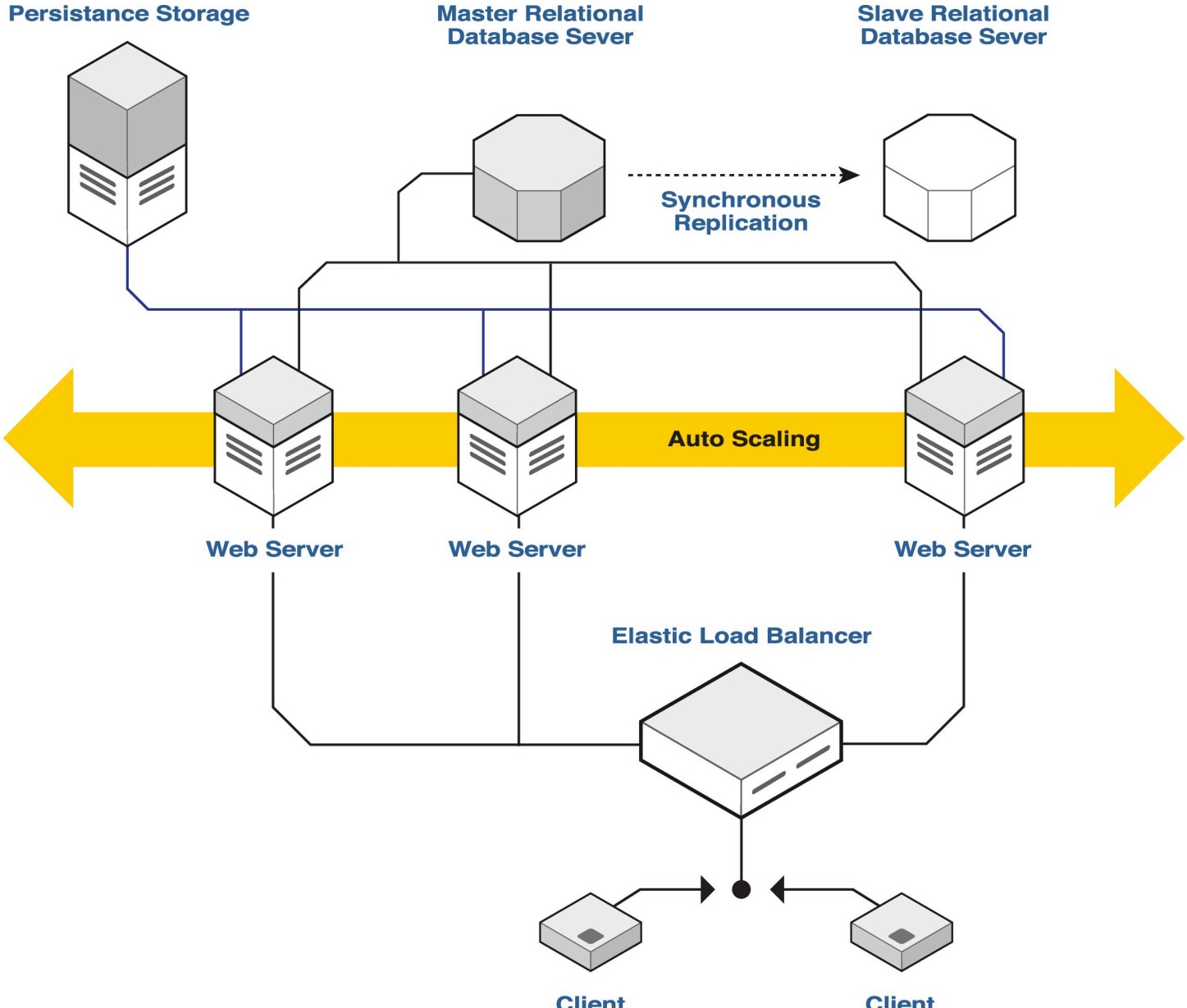

**Fig 1. Development platform infrastructure.** The control algorithm, GUI, and system engine reside in a Client machine. The Client machine communicates with the glucoregulatory model computed concurrently across several Web Servers.

traffic decreases. The ELB keeps track of active servers in a group called, "auto-scaling group" and monitors their CPU usage. As requests are made to the system, the ELB captures each request and forwards it to one of the servers in the auto-scaling group. The ELB tries to forward the requests in a fair sequential fashion without causing an imbalance in the CPU utilization among the servers in the auto-scaling group. Once the load starts to increase on the group, the ELB automatically prepares and launches into the auto-scaling group new servers with the required configuration to run the glucoregulatory models. The current protocol followed by the ELB is the following: auto-scaling group with a server minimum capacity: 1; server maximum capacity: 10; scale-up policy: step scaling when the CPU-utilization $>= 50\%$

for 1 consecutive periods of 300 seconds in the auto-scaling group; scale-down policy: simple scaling when the CPU-utilization = 0 for 1 consecutive periods of 300 seconds in the auto-scaling group; scale-up and scale-down into and out of ELB's security group; routing-algorithm: round robin; and health-check threshold: 2;

This auto-scaling feature enables high volume access to the Development Platform's cloud service. Each Web Server loads patient parameters from a secure database server that is replicated to tolerate instance failure. In addition, each Web Server is connected to a persistent storage device that loads the necessary code for running the glucoregulatory model.

## Client side

The Development Platform is designed to minimize computations on the client machine and simultaneously conceal access to the controller algorithm code. Instead of shipping code into the platform, a user can install into their code a light-weight plugin called, "Nexus Communicator" to interface with the platform. We provide the plugin for users to externally "hook up" their algorithms to the platform in Java, Python, and Matlab environments. After a user installs the Nexus Communicator, their algorithm will automatically connect to the Development Platform. Fig 2 shows an algorithm written in python code connected to the Development Platform and waiting for the patient model and clinical parameters to be set.

Fig 3 depicts the interactions of a user's code with the Nexus plugin and main application. S1 and S2 Figs present screenshots of Matlab and Java algorithms using the plugin to interface with the development platform.

The Nexus Communicator provides an object-oriented interface for algorithms to implement and for the platform to execute through secure socket programming. The Java, Python, and Matlab plugins all provide the same interface to the platform. The platform has no access to any part of the controller code and can only capture the return values of the interface methods outlined in Fig 4.

The current version of the Nexus Communicator contains two methods to provide the initial conditions set in the experiment protocol to the control algorithm, and subsequently calls

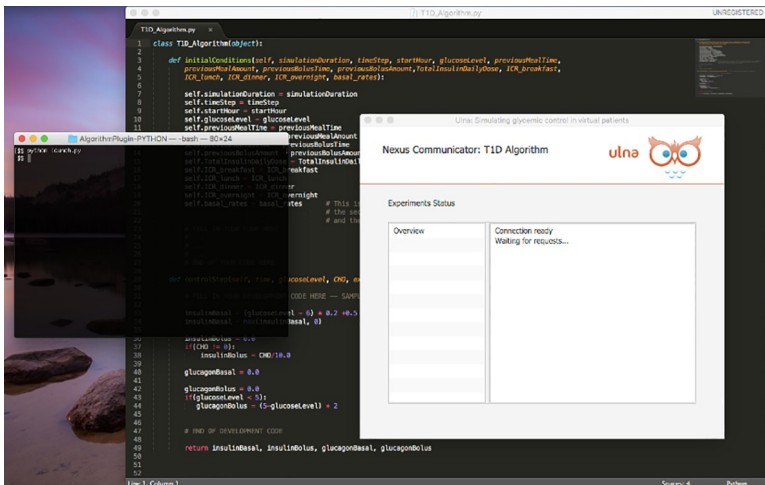

**Fig 2. Connecting a T1DM closed-loop algorithm written in python code to the development platform though the nexus communicator plugin.** The Python code implements an interface class to connect to the Platform. The plugin uses two methods to communicate insulin / glucagon dosages to the Platform and receives glucose responses and meal input. A user executes a python file that performs a system call to run the Nexus Communicator plugin. Nexus connects to the platform and controls the flow of data to instances of the python algorithm.

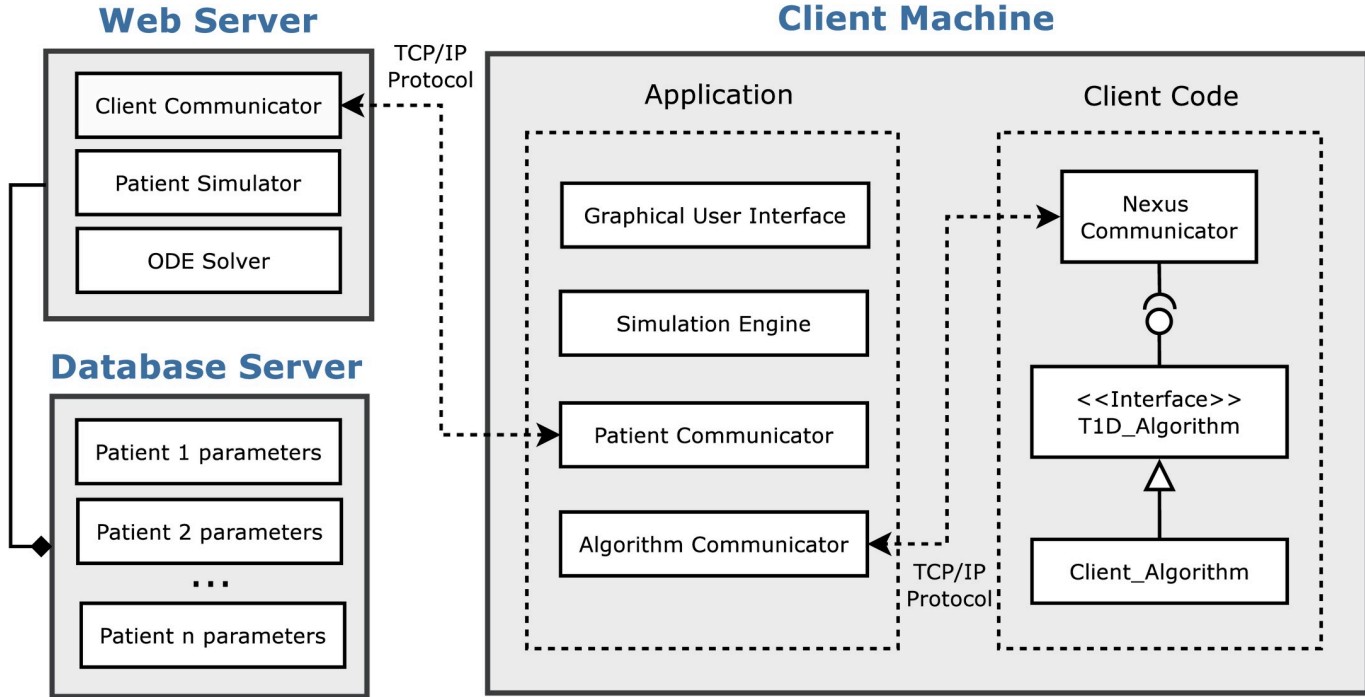

**Fig 3. Client algorithm integration into the simulation environment.** The Client Machine contains an Application and Client Code components. The Nexus Communicator is a plugin that a user can install to their controller algorithm and provides a communication interface protocol with the simulation environment.

a control Step method in a repetitive fashion until the simulation time is over. It is important to note that as the Development Platform expands and new features are added to it, the Nexus Communicator plugin interface will change to reflect the new developments. Algorithm

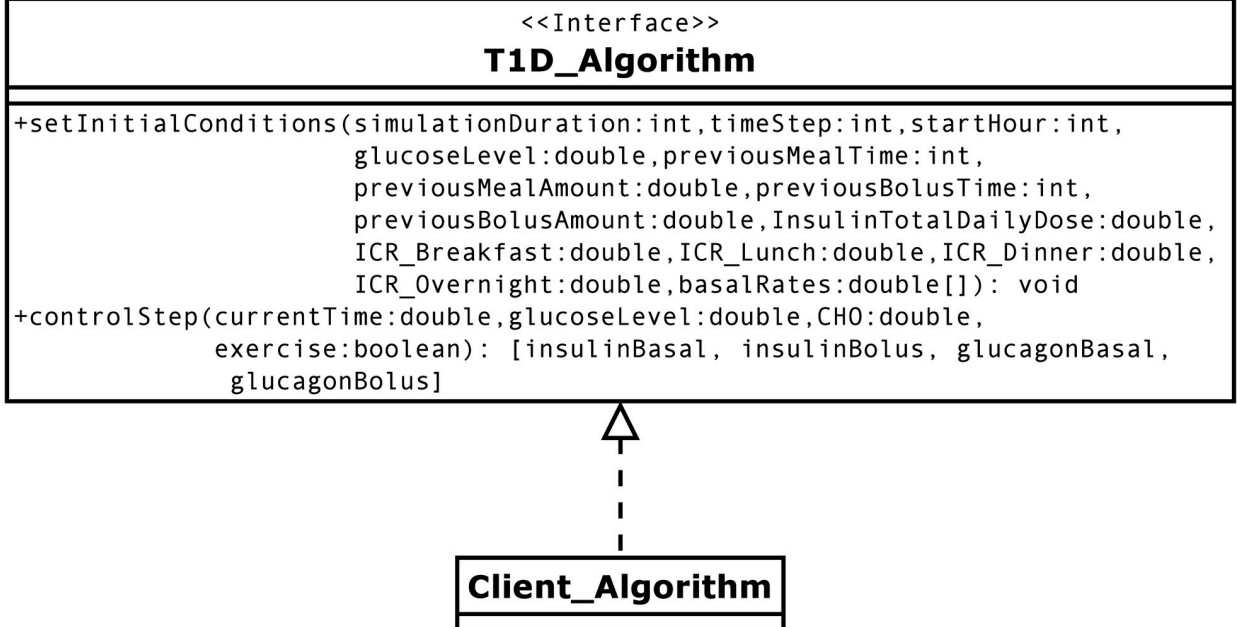

**Fig 4. Class diagram for algorithm interface implementation.** Controller algorithms implementing two basic methods can be seamlessly invoked by the simulation environment through a TCP/IP connection.

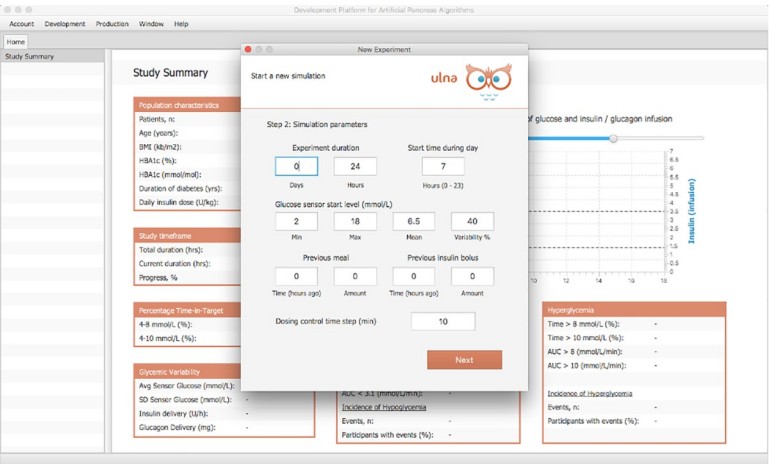

**Fig 5. Setting experiment protocol 1.** The second panel allows users to set the virtual clinical trial duration in days and hours, the start time of the trial, the variability in the starting glucose levels of the virtual subjects, the intended algorithm dosing time step.

developers should be aware of the plugin release versions they use and build their code in a modular fashion that can adapt to potential changes in the interface's method definitions.

User algorithms are completely decoupled from the development platform for code separation and security. This design allows users to easily run their algorithms against the virtual patients without having to ship code or change any of their designs. Moreover, the plugin uses object-oriented reflection to simultaneously launch multiple instances of a user's algorithm, each connected to a separate virtual patient. The plugin ensures proper thread communication over the socket and catches any exceptions that might rise in users' algorithms. Failures caused by the control algorithms do not disrupt the execution of the platform. Users can fix and re-launch their code without re-launching the platform.

The main application on the client machine also consists of a user-friendly GUI and an underlying simulation engine. The GUI is designed with the javaFX library [36] and Model-View-Controller concepts [37]. The GUI provides the user with various options to set experimental protocols. In launching a virtual clinical trial, a user can select patients from a population of virtual subjects, set the duration of the trial in days, specify any insulin onboard, set the frequency of the dosing control algorithm time-step, and define the amount of variability in glucose starting levels, meal times, meal amounts and snacks during the trial. Figs 5 and 6 present screenshots of two panels that capture this data.

The simulation engine is responsible for administering concurrent multi-threaded data communication with a user's controller algorithm, transferring controller algorithm responses to the web servers, and plotting trial outcomes to the user in real time. Fig 7 presents a sequence diagram outlining the engine's execution logic.

## Testing and debugging

Once an algorithm is run, a user can pause/resume simulations in debug mode and manually enter boluses, meals, and modify insulin-to-carbohydrate-ratios during a simulation run in real-time to test responses of their code. Users can run a step-by-step execution in real-time to follow the execution of their algorithms and check the results of their simulations, pausing at necessary occasions to check their algorithm consoles for any relevant output. An open loop / closed loop button tests the effect of switching a virtual patient back and forth to open-loop

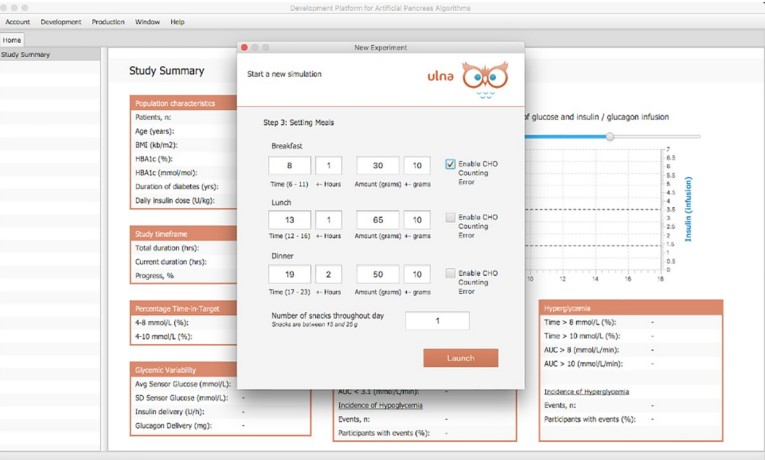

**Fig 6. Setting experiment protocol 2.** The third panel allows users to set the breakfast, lunch, dinner and snack meals times and amounts during the virtual clinical trial. The protocol allows for variability in meal amounts and times from day to day and from patient to patient. Users can enable the option of Carbohydrate Counting Error to estimate a rough 0%-20% error in insulin meal bolus calculations.

**Fig 7. Simulation engine sequence diagram.** A sequence diagram outlining the software communication protocols.

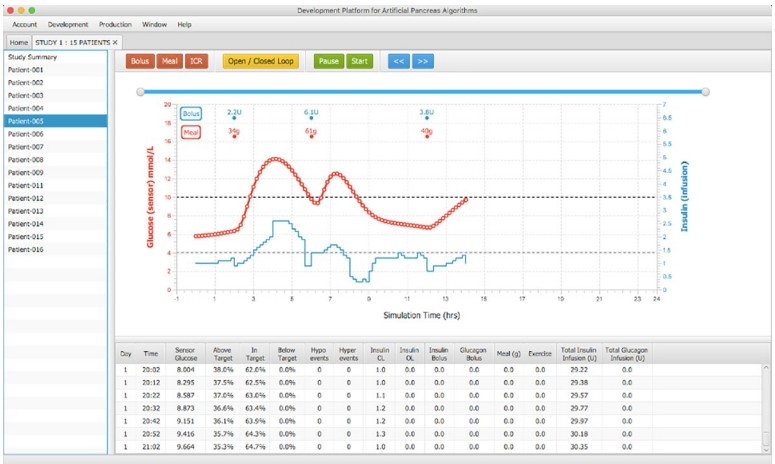

**Fig 8. Algorithm debugging mode.** Pause and Start buttons are presented in the top panel in green, open / closed Loop switch in yellow, and insulin bolus, meal CHO, and ICR buttons in red. The red line plot shows glucose readings solved by a patient model (ODE) while the blue line plots insulin values returned from a user's algorithm.

treatment, emulating the behaviour of a lost sensor connection in closed-loop or lost connection to the user algorithm. Fig 8 presents a simulation run on the Development Platform along with the debug commands available for users.

## Entire study analysis

The GUI displays to the user in real time various metrics to analyze the controller algorithm performance. The GUI graphs the glycemic responses of every patient and the overall virtual trial outcomes. The data calculated throughout the simulations include the medians, interquartile ranges, means and standard deviations of the following outcomes: percentage of Time-in-Target (4–8 mmol/L and 4–10 mmol/L), Hypoglycemia (percentage of Time < 4 mmol/L, 3.5 mmol/L, 3.1 mmol/L and area under curve AUC below 4.0 mmol/L, 3.5 mmol/L, 3.1 mmol/L), Incidence of Hypoglycemia, Hyperglycemia (percentage of Time > 8 mmol/L, 10 mmol/L and AUC above 8 mmol/L, 10 mmol/L), Incidence of Hyperglycemia, and Glycemic variability (Average Sensor Glucose mmol/L, SD Sensor Glucose mmol/L, Insulin Delivery U/h, and Glucagon Delivery mg). The population characteristics of BMI, Age, HBA1c, Daily insulin dose, and Duration of diabetes are also displayed for the chosen subjects in a particular trial. In addition to displaying the results in the GUI, the user is able export all the simulation results and virtual patient data and graphs to Microsoft Excel files. In Fig 9 we present the simulation result of 15 virtual subjects that were given 4 meals over a 24-hour period (8 am to 8 am). The first meal was given at 8 am (median 59 g of carbohydrates, IQR (40–60)), the second meal at noon (70 g, 70–75), the third meal at 5 pm (95 g, 81–100), and a snack at 9 pm (20 g, 20–30).

## Modeling virtual patients

The Development Platform includes a program to simulate and run virtual subjects. Interested developers who want to test algorithms against specific patient models of their choice can implement their own versions of virtual subjects on the platform. The virtual subject class can be extended in an object-oriented manner and new implementations of virtual patients can be achieved using Java. For developers only interested in creating algorithms, we provide in the Development Platform a glucoregulatory system of thirteen ODEs to model virtual subjects.

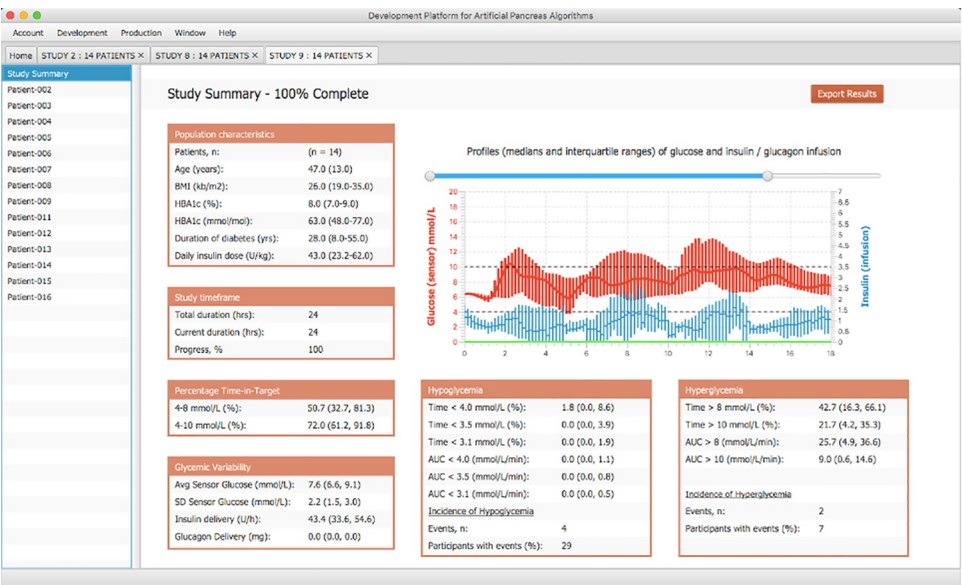

**Fig 9. Algorithm performance report assessing the clinical outcomes of a 24-hour single-hormone treatment of a single-hormone treatment.** Results include baseline characteristics of study participants, percentage time in target, glycemic variability, hypoglycemia and hyperglycemia analyses.

The default glycemic responses of virtual subjects are defined by a unique set of parameters that simulate inter and intra subject variability.

The default model uses a nonlinear model of glucose and a stochastic estimation method to determine virtual subject parameters. The glucoregulatory model consists of seven subsystems that model the following dynamics: subcutaneous insulin absorption, plasma insulin kinetics, insulin action, glucagon absorption dynamics, gut glucose absorption, plasma glucose dynamics, and interstitial glucose dynamics. For each component, several models have been proposed in the literature [38]. Input to the model include subcutaneous insulin infusion, subcutaneous glucagon infusion, and carbohydrate content of meals. The main output we capture from the model is the plasma glucose concentration. Below are short descriptions of the subsystems.

### Insulin absorption dynamics (pharmacokinetics)

The absorption of subcutaneously-infused insulin can be modelled as two parallel absorption channels, fast and slow [39]. The slow channel comprises a two-compartment chain with two different transfer rates while the fast channel comprises a two-compartment chain with one shared transfer rate. The slow channel is described by the following equations:

$$\dot{Q}_{is1}(t) = u_i(t)p_i - Q_{is1}(t)k_{is1}, \tag{1}$$

$$\dot{Q}_{is2}(t) = Q_{is1}(t)k_{is1} - Q_{is2}(t)k_{is2}, \tag{2}$$

where $Q_{is1}(t)$ and $Q_{is2}(t)$ (U) are the insulin masses in the first and second compartment, respectively, $u_i(t)$ (U/min) is the insulin infusion rate, $p_i$ (unitless) is the portion of subcutaneous insulin that is absorbed through the slow channel, and $k_{is1}$ and $k_{is2}$ (1/min) are the fractional transfer rate parameters. The fast channel is described by the following equations:

$$\dot{Q}_{if1}(t) = u_i(t)(1 - p_i) - Q_{if1}(t)k_{if}, \tag{3}$$

$$\dot{Q}_{if2}(t) = Q_{if1}(t)k_{if} - Q_{if2}(t)k_{if}, \tag{4}$$

where $Q_{if1}(t)$ and $Q_{if2}(t)$ (U) are the insulin masses in the first and second compartment, respectively, and $k_{if}$ (1/min) is the shared fractional transfer rate.

## Plasma insulin kinetics

We represent plasma insulin kinetics by one compartment

$$\dot{Q}_i(t) = (Q_{is2}(t)k_{is2} + Q_{if2}(t)k_{if})I_m(t) - Q_i(t)k_e + c_i, \tag{5}$$

where $Q_i(t)$ (U) is the insulin mass in the plasma, $k_e$ (1/min) is the fractional clearance rate, $c_i$ (U/min) is the background insulin appearance, and $I_m(t)$ (unitless) is a multiplicative time-varying piecewise-linear function describing diurnal and other time-varying components of insulin kinetics. The plasma insulin concentration $I_p(t)$ (mU/l) is obtained as

$$I_p(t) = \frac{Q_i(t)}{V_i w} x\, 10^6 \tag{6}$$

where $V_i = 190$ ml/kg is the insulin distribution volume and $w$ (kg) is subject's weight.

## Insulin action dynamics

Insulin action can be partitioned into suppression of endogenous glucose production, promotion of glucose disposal, and distribution of glucose [40]. The effect of insulin lags behind plasma insulin concentrations by 10–30 minutes [41], and thus, state variables of the delayed effect (referred to as remote effect) of insulin are defined as:

$$\dot{x}_1(t) = -k_{a1}x_1(t) + k_{a1}I_p(t), \tag{7}$$

$$\dot{x}_2(t) = -k_{a2}x_2(t) + k_{a2}I_p(t), \tag{8}$$

$$\dot{x}_3(t) = -k_{a3}x_3(t) + k_{a3}I_p(t), \tag{9}$$

where $x_1(t)$, $x_2(t)$, and $x_3(t)$ (1/min) are the delayed effects of insulin on glucose distribution, glucose disposal, and the endogenous glucose production, respectively, $k_{a1}$, $k_{a2}$, and $k_{a3}$ (1/min) are time constants.

## Glucagon absorption dynamics (pharmacokinetics)

The absorption of subcutaneously-infused glucagon can be modelled as [42]:

$$\dot{Q}_{g1}(t) = u_g(t) - \frac{Q_{g1}(t)}{t_{g\,max}}, \tag{10}$$

$$\dot{Q}_{g2}(t) = \frac{Q_{g1}(t)}{t_{g\,max}} - \frac{Q_{g2}(t)}{t_{g\,max}}, \tag{11}$$

where $Q_{g1}(t)$ and $Q_{g2}(t)$ (units) are the glucagon masses in the first and second subcutaneous compartments, respectively, $u_g(t)$ (unit/min) is the glucagon infusion rate, and $t_{g\,max}$ (min) is time-to-peak plasma glucagon concentration following a glucagon bolus (impulse response).

The plasma glucagon concentration $C_p(t)$ is obtained as:

$$C_p(t) = \frac{1}{t_{g_{max}}} \cdot \frac{Q_{g2}(t)}{w \cdot MCR_g} \times 10^6 + C_b, \tag{12}$$

where $MCR_g$ (ml/kg/min) is the metabolic clearance rate and $C_b$ (mU/l) is the background plasma glucagon concentration.

## Meal absorption dynamics

Two absorption channels can represent the absorption of meal glucose from the gut, each comprising a two-compartment chain to allow double-peak absorption profiles to be represented. Double-peak absorption profiles were observed in a previous study [43]. Absorption channels share the fractional transfer rate. To allow increased flexibility of absorption profiles that considerably vary between subjects, the model includes a multiplicative time-varying function. The subsystem is described as:

$$U_m(t) = (U_{m1}(t) + U_{m2}(t))f_m(t) \tag{13}$$

$$U_{m1}(t) = k_m^2 t e^{-k_m t} \frac{CHOx5551}{w} p_m \tag{14}$$

$$U_{m2}(t) = \begin{cases} k_m^2(t-d)e^{-k_m(t-d)} \frac{CHOx5551}{w}(1-p_m) & \text{if } t > d \\ 0 & \text{otherwise} \end{cases}, \tag{15}$$

where $U_{m1}(t)$ and $U_{m2}(t)$ (μmol/kg/min) are the appearances of the first and the second absorption channels, respectively, $k_m(t)$ (1/min) is a transfer rate parameter, $d$ (min) is the delay associated with the second absorption channel, $p_m$ (unitless) is the portion of meal carbohydrates absorbed through the first channel, $CHO$ (g) is the carbohydrate content of the meal, and $f_m(t)$ (unitless) is the time-varying piecewise-linear function.

## Plasma glucose dynamics

Glucose kinetics are represented by model developed in [40], adapted to allow glucagon action [44]:

$$\dot{Q}_1(t) = -F_{01} \frac{\frac{Q_1(t)}{160}}{1 + \frac{Q_1(t)}{160}} - x_1(t)S_t Q_1(t) + k_{12}Q_2(t) + EGP(t) + F_g(t) + U_m(t), \tag{16}$$

$$\dot{Q}_2(t) = x_1(t)S_t Q_1(t) - (k_{12} + x_2(t)S_d)Q_2(t), \tag{17}$$

$$EGP(t) = \begin{cases} C_p(t)S_g(1 - x_3(t)S_e) & \text{if } x_3(t) < 1 \\ 0 & \text{if } x_3(t) \geq 1 \end{cases}, \tag{18}$$

where $Q_1(t)$ and $Q_2(t)$ (μmol/kg) are the glucose masses in the accessible (where measurements are made) and the non-accessible glucose compartments, respectively, $F_{01}$ (μmol/kg/min) is the non-insulin dependent glucose utilization, and $S_g$ is the glucagon sensitivity, $S_t$ and $S_d$ ($10^{-4} \times$ /min per mU/l) are the insulin sensitivities of glucose distribution and glucose disposal, respectively, and $S_e$ ($10^{-4}$ per mU/l) is the insulin sensitivity of endogenous glucose production, and $F_g(t)$ (unitless) is a time-varying piecewise-linear flux function capturing diurnal

changes and other time-varying characteristics of glucose dynamics. The plasma glucose concentration $G(t)$ (mmol/L) is obtained as:

$$G(t) = \frac{Q_1(t)}{V},$$
(19)

where $V$ is the glucose distribution volume.

### Interstitial glucose dynamics

Interstitial glucose dynamics can be represented by:

$$\dot{G}_s(t) = -k_s G_s(t) + k_s G(t),$$
(20)

where $G_s(t)$ is the interstitial glucose concentration (where the sensor measures) and $k_s$ (1/min) is a time constant. The model incorporates time-invariant parameters and time-variant functions (fluxes) which are used to outline insulin changes and glucose responses.

Virtual subjects are represented by a set of parameter vectors. Values of time-variant and time-invariant parameters were obtained using the method described in [39]. The system currently includes 15 virtual subjects that each represents a real patient that was tested in a clinical trial [45].

## Results & discussion

The focus of the platform's design is the separation of the three programs running the virtual patients, the GUI, and the user algorithm. This separation ensures complete isolation of user algorithms from virtual patients and helps distribute the load of computations over a cloud network. The servers running on the cloud network are CPU-optimized to solve the ODE system as efficiently as possible. The available servers each have between 16 and 48 CPU cores and implement hyper-threading to provide 32–96 vCPUs each. Consequently, with this separation, a user's machine only runs their control algorithm and waits for the solution to each ODE system at every iteration. In Fig 10 we plot the results of running an AP virtual clinical trial of 16 patients using a dual-hormone algorithm on an i3 dual-core machine (4 vCPUs) and an i7 quad-core machine (8 vCPUs) supporting hyper-threading for 60 simulation days. For each machine, we ran the 16 patients simultaneously using the Development Platform over the cloud (distributed) versus a complete local installation (sequential) where the ODE

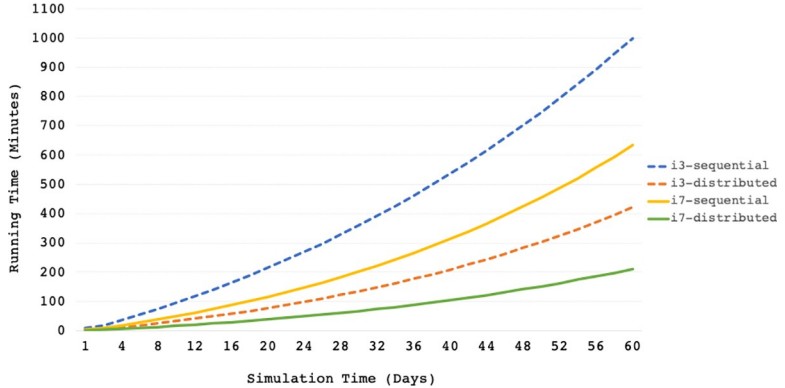

**Fig 10. Computational burden of simulating a virtual clinical trial of 16 patients.** A comparison between Core i3 and Core i7 processors running 16 virtual patient processes simultaneously using the Development Platform over the cloud (distributed) versus a complete local installation on a physical device (sequential).

model was solved on the same machine. The distributed simulation over the cloud provided a 62% improvement in the running time of the virtual clinical trial on the i3 machine and a 66% improvement on the i7 machine. The distributed simulation on the i3 processor with half the computational power of the i7 processor still ran 33% faster than the sequential simulation on the i7 processor. This overall improvement in speed and CPU utilization can help users develop their algorithms twice as fast and run longer simulations. The reduction in the computational burden of virtually running an AP system opens the possibilities to develop learning algorithms that can train on very long simulation durations [46–49] efficiently without requiring users to obtain specialized hardware.

Using the glucose and insulin readings, meal data, and experimental protocols of one of our previous dual-hormone randomized crossover controlled trials [45], we ran Bayesian Markov chain Monte Carlo (MCMC) simulations using WinBUGS [50]. We ran the virtual patient program over the cloud platform and used the Nexus Communicator to plug in the Matlab algorithm that was used to conduct the CLASS03 clinical study [8]. We mimicked the protocol of the CLASS03 trial and used the Development Platform panels shown in S4 and S5 Figs, 5 and 6 to set a simulation timeframe of 24-hours (8 am to 8 am) and 4 meals to be given to each of the 15 virtual patients. We set the first meal at 8 am (median 59 g of carbohydrates, IQR (40–60)), the second meal at noon (70 g, 70–75), the third meal at 5 pm (95 g, 81–100), and a snack at 9 pm (20 g, 20–30). The patients in CLASS03 participated in an open-loop treatment, a single-hormone treatment, and a dual-hormone treatment on separate occasions. We replicated all three treatments on our set of virtual patients using the same open-loop and closed-loop algorithm used in the trial. The simulations all ran in parallel for each treatment, launching 15 instances of each algorithm all running simultaneously with each connected to a single patient. On a personal machine, the entire simulation took less than 20 minutes to setup, run, and terminate. S6 and S9 Figs, and 9 present a visual report generated by the development platform for the virtual clinical trials. S7, S8 and S10 Figs graph the response on the platform of one of the virtual patients to the different algorithms used during the simulation run.

We report in S1 Table the baseline characteristics for the virtual patients and the patients in CLASS03. The virtual patients' mean age and duration of diabetes are slightly higher than the patients in CLASS03 (47 vs 33 and 28 vs 16, respectively). However, the BMI, HbA1c, and total daily dose are within close range. S2 Table outlines the main clinical results of the virtual trial, comparing the performance of open-loop, single-hormone, and dual-hormone treatments. Table 1 compares the outcomes of the virtual trials to CLASS03. The time In target, time between 4.0–10.0 mmol/L, time above 8.0 mmol/L, time above 10.0 mmol/L, and plasma glucose concentrations are all nearly identical in both studies for the three arms. Moreover, the mean glucose levels, SD of glucose, insulin delivery, and glucagon delivery were extremely close in both studies for the three arms. We report in Table 1 the means and medians of the times below 4.0 mmol/L. Patients in CLASS03 spent slightly more time in hypoglycemia during the day due to a 30-minute exercise they performed in the study. The 30-minute exercise also lead to more hypoglycemic events (Table 2) than was observed in the virtual trial. Simulating exercise is something we plan to integrate into the default virtual patients on development platform in the future. Nevertheless, the rates of hypoglycemia in both studies followed the same decrease from open-loop to single-hormone to dual-hormone treatments. In addition, overnight outcomes reported in S3 Table also confirm very similar outcomes between the virtual and CLASS03 trial. The paired difference between the three treatments in the virtual trial and CLASS03 are also extremely similar (comparing S2 Table with data from CLASS03).

**Table 1. Comparisons between simulated and real experiments using 24-hour clinical data collected in the CLASS03 randomized trial involving the dual-hormone artificial pancreas, single-hormone artificial pancreas, and conventional insulin pump therapy arms.**

| Outcome | Conventional insulin pump therapy | | Single-hormone artificial pancreas | | Dual-hormone artificial pancreas | |
|---|---|---|---|---|---|---|
| | CLASS03 (n = 29) | Simulations (n = 14) | CLASS03 (n = 30) | Simulations (n = 14) | CLASS03 (n = 29) | Simulations (n = 14) |
| Time spent at glucose levels (%): | | | | | | |
| In target* | 51% (19) | 52% (25) | 62% (18) | 63% (20) | 63% (18) | 64% (25) |
| 4.0–10.0 mmol/L | 61% (20) | 63% (19) | 74% (15) | 74% (17) | 77% (14) | 74% (22) |
| < 4.0 mmol/L | 13.1% (11.6), 13.3% (2.8–20.9) | 9.1% (11.5), 1.7% (0.0–17.5) | 5.7% (6.8), 3.1% (0.6–8.7) | 4.3% (5.3), 1.7% (0.0–7.9) | 2.7% (3.5), 1.5% (0.0–3.5) | 1.2% (2.9), 0.0% (0.0–0.0) |
| < 3.5 mmol/L | 6.6% (6.1), 4.8% (1.4–10.5) | 5.9% (8.5), 0.0% (0.0–11.0) | 3.0% (4.6), 0.0% (0.0–4.8) | 1.9% (3.0), 0.0% (0.0–3.3) | 0.9% (1.5), 0.1% (0.0–1.4) | 0.5% (1.4), 0.0% (0.0–0.0) |
| < 3.3 mmol/L | 4.3% (4.3), 3.0% (1.1–6.5) | 4.7% (6.7), 0.0% (0.0–8.7) | 1.8% (3.3), 0.0% (0.0–2.0) | 1.4% (2.3), 0.0% (0.0–2.1) | 0.6% (1.2), 0.0% (0.0–0.8) | 0.3% (0.9), 0.0% (0.0–0.0) |
| > 8.0 mmol/L | 41% (23) | 45% (33) | 39% (19) | 40% (25) | 41% (18) | 42% (26) |
| > 10.0 mmol/L | 26% (22) | 28% (26) | 20% (16) | 22% (19) | 20% (15) | 24% (23) |
| Mean glucose (mmol/L) | 7.9 (2.1) | 8.0 (2.5) | 7.7 (1.4) | 7.9 (1.7) | 8.0 (1.4) | 8.3 (1.9) |
| SD of glucose (mmol/L) | 2.9 (0.9) | 2.5 (0.8) | 2.6 (0.9) | 2.4 (1.1) | 2.5 (0.9) | 2.3 (1.1) |
| Insulin delivery (U/kg) | 0.69 (0.18) | 1.1 (0.38) | 0.63 (0.17) | 1.0 (0.4) | 0.62 (0.19) | 1.1 (0.44) |
| Glucagon delivery (mg) | - | - | - | - | 0.04 (0.02–0.08)^ | 0.03 (0.0–0.12) |

Data are mean (SD) or median (IQR), unless otherwise indicated.

*Primary outcome, defined as 4·0–10·0 mmol/L for 2 h postprandially and 4·0–8·0 mmol/L otherwise.

^Glucagon delivery for day time period outside exercise range.

## Conclusion & future work

Conducting clinical trials to develop early stage algorithms, to test new improvements, to test multiple treatments [51], or to validate algorithm performance under varying conditions is expensive and can be facilitated by a computational platform that conducts virtual clinical trials. Although not a replacement for clinical studies, such a platform would considerably save resources, time, and speed development of AP systems for T1DM patients.

The conclusions reached in the virtual trial are the same clinical conclusions observed in CLASS03 regarding the performance of the open-loop, single-hormone, and dual-hormone

**Table 2. Hypoglycaemia rates and nocturnal hypoglycaemia.**

| Outcome | Conventional insulin pump therapy | | Single-hormone artificial pancreas | | Dual-hormone artificial pancreas | |
|---|---|---|---|---|---|---|
| | CLASS03 (n = 29) | Simulations (n = 14) | CLASS03 (n = 29) | Simulations (n = 14) | CLASS03 (n = 29) | Simulations (n = 14) |
| Number of hypoglycemic events | 52 | 11 | 13 | 4 | 9 | 3 |
| Number of nocturnal hypoglycemic events | 13 | 4 | 0 | 0 | 0 | 0 |
| Number of patients with at least one hypoglycemia event | 24 (83%) | 6 (43%) | 5 (17%) | 4 (29%) | 6 (21%) | 2 (14%) |

Data are number (%) or numbers. Hypoglycaemic events for CLASS03 are defined as plasma glucose concentration below 3·3 mmol/L with symptoms or below 3·0 mmol/L irrespective of symptoms, and were treated with oral carbohydrate. Hypoglycaemic events for Simulations are defined as glucose concentration below 3·3 mmol/L and were treated with simulated oral carbohydrate.

algorithms used. The results suggest that the default T1DM virtual subjects we provide in the Development Platform are a good representation of the T1DM population who participated in the clinical study. The development platform is intended to give algorithm developers a venue to efficiently develop and rapidly test their T1DM solutions prior to clinical testing.

Changes and updates to the virtual patient program running on the cloud can be made without requiring users to download updates. The separation of concerns in the platform allows for the ODE model to be updated or changed without affecting users' access to the platform. Moreover, users have automatic access to new virtual patients as they become available on the platform's database, without the need to run updates. However, new additions to the core of the plugin will require users to update their codes to match any interface changes.

To help users better assess the performance of their algorithms and how they compare to open-loop treatment, the platform provides a feature that runs a default open-loop virtual trial with user defined protocols. Users can compare the performance of their algorithms to open-loop running on the same protocols. The platform maintains meal rescue protocols for open-loop using procedures defined by the Medtronic Minimed Pump manual. We plan to add other protocols and procedures by other companies in future releases. The platform does not currently integrate directly into any of the CGMs or pumps for testing. Nevertheless, its current architectures allows for many commercial features to be integrated as the platform grows. It is designed to assist interested researchers in trying new ideas and approaches that could potentially motivate new advances in commercial algorithms.

An advantage of designing the development platform with object-oriented principles is the compartmentalization of the different system components. A component such as the ODE model can be easily replaced with another model without rewriting or restructuring existing code. The platform can support different model structures and different parameter values without the need for the user to do anything locally. Future updates to the platform would include the addition of more virtual patients, the extension of the existing default model to incorporate exercise modeling, the modeling of more complex sensor dynamics, the modeling of flux variability from day to day, and the ability for users to create custom ODE models and upload their own patient parameters. Access to the development platform is made possible through a desktop application that can be downloaded at http://t1dclinic.com/ulna.php.

## Supporting information

**S1 Material.**
(PDF)

## Author Contributions

**Conceptualization:** Mohamed Raef Smaoui, Remi Rabasa-Lhoret, Ahmad Haidar.

**Data curation:** Mohamed Raef Smaoui, Remi Rabasa-Lhoret, Ahmad Haidar.

**Formal analysis:** Mohamed Raef Smaoui, Remi Rabasa-Lhoret, Ahmad Haidar.

**Investigation:** Mohamed Raef Smaoui, Remi Rabasa-Lhoret, Ahmad Haidar.

**Methodology:** Mohamed Raef Smaoui, Ahmad Haidar.

**Project administration:** Mohamed Raef Smaoui, Remi Rabasa-Lhoret, Ahmad Haidar.

**Resources:** Mohamed Raef Smaoui, Remi Rabasa-Lhoret, Ahmad Haidar.

**Software:** Mohamed Raef Smaoui.

**Supervision:** Mohamed Raef Smaoui, Remi Rabasa-Lhoret.

**Validation:** Mohamed Raef Smaoui, Ahmad Haidar.

**Visualization:** Mohamed Raef Smaoui.

**Writing – original draft:** Mohamed Raef Smaoui.

**Writing – review & editing:** Mohamed Raef Smaoui, Remi Rabasa-Lhoret, Ahmad Haidar.

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
