## [Decision Letter · Decision Letter 0]

2 Nov 2020

PONE-D-20-25426

Development Platform for Artificial Pancreas Algorithms

PLOS ONE

Dear Dr. Smaoui,

Thank you for submitting your manuscript to PLOS ONE. After careful consideration, we feel that it has merit but does not fully meet PLOS ONE’s publication criteria as it currently stands. Therefore, we invite you to submit a revised version of the manuscript that addresses the points raised during the review process.

We look forward to receiving your revised manuscript.

Kind regards,

Othmar Moser

Academic Editor

PLOS ONE

Journal Requirements:

2. Please clearly report at the beginning of your methods or results section which the key performance measures were to establish validity and utility of your model/algorithm/tool.

Reviewers' comments:

Reviewer's Responses to Questions

**Comments to the Author**

1. Is the manuscript technically sound, and do the data support the conclusions?

Reviewer #1: Yes

Reviewer #2: Partly

Reviewer #3: Yes

2. Has the statistical analysis been performed appropriately and rigorously? 

Reviewer #1: Yes

Reviewer #2: I Don't Know

Reviewer #3: I Don't Know

3. Have the authors made all data underlying the findings in their manuscript fully available?

Reviewer #1: Yes

Reviewer #2: Yes

Reviewer #3: Yes

4. Is the manuscript presented in an intelligible fashion and written in standard English?

Reviewer #1: Yes

Reviewer #2: Yes

Reviewer #3: Yes

5. Review Comments to the Author

Reviewer #1: I'd like to applaud the authors for this publication. The development of AP systems is one of the most important topics in research around type 1 diabetes. Even though this publication solely included virtual patients, the level of detail that has been provided regarding the methodology, structure, development and user friendly application within this manuscript is unusual and absolutely outstanding.

Furthermore, the writing is, even though the topic is rather complicated, exceptional and enjoyable.

This makes me recommend this manuscript for publication without any further comments, since individuals that are interested in this specific topic will gain valuable insight into an otherwise unexplored field of research.

Reviewer #2: This paper describes a science or IT topic and emphasizes on a statistical issue: the authors talk about a clinically validated cloud-based distributed platform that supports the development and comprehensive testing of single and dual-hormone algorithms for type 1 diabetes mellitus (T1DM). The platform utilizes the validated patient model to conduct virtual clinical trials for the rapid development and testing of closed-loop algorithms for T1DM.

From a medical point of view one can assume that the need for virtual clinical trials is low. We need clinical trials on patients with type 1 diabetes to find out more about their glucose metabolism. If virtual clinical trials fit the real needs remains to be proven.

A reviewer with strong IT skills combined with statistical knowledge (best fit with knowledge of artificial intelligence) should review this paper.

Reviewer #3: The content of this manuscript is highly important in order standardize and facilitate the evaluation of the performance of AP systems. In general, the article is well written and the aim is clear. I am not able to prove the mathematical and statistical correctness of the models. I suggest them to be rechecked by a specialized person.

Further minor points:

- I would recommend not to use the term “patient” for users of AP technology throughout the manuscript

- Line 90: “algorithms must prove effective clinically in…” please check the grammar

6. PLOS authors have the option to publish the peer review history of their article (what does this mean?). If published, this will include your full peer review and any attached files.

Reviewer #1: No

Reviewer #2: No

Reviewer #3: No

---

## [Author Response · Author response to Decision Letter 0]

9 Nov 2020

Reviewer 1:

We sincerely thank the reviewer for their time in reading the manuscript and recommending it for publication. We are delighted that the reviewer has enjoyed reviewing the work.

Reviewer 2:

We thank the reviewer for taking the time to read and review the manuscript. We agree with the reviewer that more clinical trials on patients with type 1 diabetes are needed to better understand glucose metabolism. Data from future clinical trials in the field will continue to improve our patient models and provide better testing frameworks for algorithms. Nonetheless, the community of AP algorithm developers can still greatly benefit from the currently existing clinical data and models provided by a platform such as the one we are presenting to test and improve algorithms.

Reviewer 3:

We thank the reviewer for their time and effort in reviewing the manuscript and are delighted that the reviewer found the topic of high importance. We are confident of the correctness of the mathematical and statistical models.

Concern:

I would recommend not to use the term “patient” for users of AP technology throughout the manuscript.

Response:

We have reworded most of the phrases that referred to users of AP technology as “patients”. We also replaced the term with “participants” / “subjects” / “users” / “humans” throughout the manuscript wherever it seemed appropriate.

Concern:

Line 90: “algorithms must prove effective clinically in..” please check the grammar.

Response:

We thank the reviewer for pointing this out. We have rephrased the sentence to fix the grammar mistake.

---

## [Decision Letter · Decision Letter 1]

17 Nov 2020

Development Platform for Artificial Pancreas Algorithms

PONE-D-20-25426R1

Dear Dr. Smaoui,

We’re pleased to inform you that your manuscript has been judged scientifically suitable for publication and will be formally accepted for publication once it meets all outstanding technical requirements.

Kind regards,

Othmar Moser

Academic Editor

PLOS ONE

Reviewers' comments:

Reviewer's Responses to Questions

**Comments to the Author**

1. If the authors have adequately addressed your comments raised in a previous round of review and you feel that this manuscript is now acceptable for publication, you may indicate that here to bypass the “Comments to the Author” section, enter your conflict of interest statement in the “Confidential to Editor” section, and submit your "Accept" recommendation.

Reviewer #1: All comments have been addressed

2. Is the manuscript technically sound, and do the data support the conclusions?

Reviewer #1: Yes

3. Has the statistical analysis been performed appropriately and rigorously? 

Reviewer #1: I Don't Know

4. Have the authors made all data underlying the findings in their manuscript fully available?

Reviewer #1: Yes

5. Is the manuscript presented in an intelligible fashion and written in standard English?

Reviewer #1: Yes

6. Review Comments to the Author

Reviewer #1: (No Response)

7. PLOS authors have the option to publish the peer review history of their article (what does this mean?). If published, this will include your full peer review and any attached files.

Reviewer #1: No

---

## [Editor Report · Acceptance letter]

23 Nov 2020

PONE-D-20-25426R1 

Development Platform for Artificial Pancreas Algorithms 

Dear Dr. Smaoui:

I'm pleased to inform you that your manuscript has been deemed suitable for publication in PLOS ONE. Congratulations! Your manuscript is now with our production department. 

Kind regards, 

on behalf of

Dr. Othmar Moser 

Academic Editor

PLOS ONE